# CFRWD-GAN for SAR-to-Optical Image Translation

**Juan Wei, Huanxin Zou \* , Li Sun, Xu Cao, Shitian He, Shuo Liu and Yuqing Zhang**

College of Electronic Science and Technology, National University of Defense Technology, Changsha 410073, China; weijuan20@nudt.edu.cn (J.W.); sunli20@nudt.edu.cn (L.S.); cx2020@nudt.edu.cn (X.C.); heshitian19@nudt.edu.cn (S.H.); liushuo21@nudt.edu.cn (S.L.); zhangyuqing22@nudt.edu.cn (Y.Z.)
\* Correspondence: zouhuanxin@nudt.edu.cn; Tel.: +86-731-8700-3288

**Abstract:** Synthetic aperture radar (SAR) images have been extensively used in earthquake monitoring, resource survey, agricultural forecasting, etc. However, it is a challenge to interpret SAR images with severe speckle noise and geometric deformation due to the nature of radar imaging. The translation of SAR-to-optical images provides new support for the interpretation of SAR images. Most of the existing translation networks, which are based on generative adversarial networks (GANs), are vulnerable to part information loss during the feature reasoning stage, making the outline of the translated images blurred and semantic information missing. Aiming to solve these problems, cross-fusion reasoning and wavelet decomposition GAN (CFRWD-GAN) is proposed to preserve structural details and enhance high-frequency band information. Specifically, the cross-fusion reasoning (CFR) structure is proposed to preserve high-resolution, detailed features and low-resolution semantic features in the whole process of feature reasoning. Moreover, the discrete wavelet decomposition (WD) method is adopted to handle the speckle noise in SAR images and achieve the translation of high-frequency components. Finally, the WD branch is integrated with the CFR branch through an adaptive parameter learning method to translate SAR images to optical ones. Extensive experiments conducted on two publicly available datasets, QXS-SAROPT and SEN1-2, demonstrate a better translation performance of the proposed CFRWD-GAN compared to five other state-of-the-art models.

**Keywords:** SAR-to-optical image translation; generative adversarial networks; cross-fusion reasoning structure; wavelet decomposition

## 1. Introduction

For several decades, remote sensing technology has become a very advanced space exploration technology [1–3]. Among them, synthetic aperture radar (SAR) and optical remote sensing devices are broadly applied in land planning, disaster prevention, target detection, and other aspects [4–7]. However, the harsh environment and light changes cause great interference to the optical remote sensing sensor and seriously affect ground observation. SAR can provide high-resolution imaging in various weather conditions, providing supplementary information for optical remote sensing images. Unfortunately, due to the imaging characteristics of SAR images, their interpretation has been a bottleneck. First, due to a large amount of speckle noise generated by the coherent interference of the target scattering radar echo, it is difficult to obtain effective information in SAR images. Second, the SAR signal wavelength (mm~cm) fails to represent the observable part of the familiar electromagnetic band perceived by the human eye. The comprehensible features in the optical images may significantly be confused in SAR images [8]. Figure 1a,b represents a SAR image taken by the Sentinel-1 satellite and an optical image taken by the Sentinel-2 satellite in the same area, respectively. It can be seen that the middle white river and the surrounding green vegetation are clearly distinguishable in the optical image, while they are nonintuitive in the SAR image, with extensive speckle noise. As many orbit radar satellites obtain abundant radar images, it is urgent to develop an efficient interpretation method.

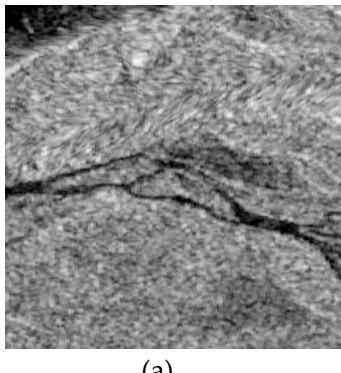 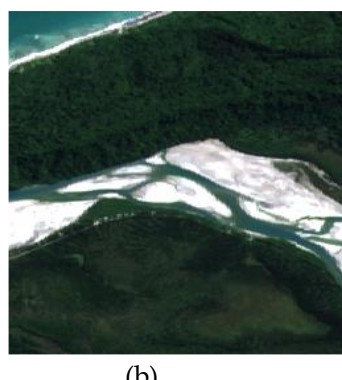

<center>(a)                              (b)</center>

**Figure 1.** A contrast between SAR and optical images taken in the same area around 1 March 2017 to 30 May 2017 [9]. (**a**) A SAR image taken by Sentinel-1 satellite. (**b**) An optical image taken by Sentinel-2 satellite.

Over the past years, the interpretation of SAR images has evolved from manual to semi-automatic and is currently trending towards automation and intelligence. With the advance of deep learning (DL) technologies, especially the proposal of GANs [10], it is possible to learn the translation between images in different domains. GANs are a powerful tool for generating images under a given condition. In the field of SAR-to-optical image translation (STOIT), there are many methods based on GANs, such as pix2pix [11] and cycleGAN [12]. In recent years, many scholars have improved GAN-based networks [13]. Nevertheless, these methods are mainly transferred from general image-to-image translations (ITIT), which may not be optimal for converting SAR images to optical images and may result in considerable information loss. Moreover, the speckle noise of SAR images also has a large impact on the translation performance.

Driven by the fact that detailed information is lost heavily in the feature reasoning stage, this paper proposes a novel structure of feature reasoning. Unlike the reasoning structures commonly used in the pix2pix and cycleGAN, the residual blocks and parallel structures are combined in our work to construct multi-scale branches and thus complete the reasoning process, where different scale features are linked together by cross-fusion. Additionally, the wavelet decomposition (WD) branch is designed to decrease the speckle noise in SAR images and recover high-frequency details. The output from the branch with CFR structure is fused with the WD branch output through a learnable coefficient. The key contributions are concluded as follows:

1. A CFR structure is proposed for converting SAR to optical images. Compared with the U-Net and the cascade of nine residual blocks (CN-ResBlocks) reasoning structures in the pix2pix and the cycleGAN, the proposed CFR structure can extract and retain more information.
2. A WD branch is developed to retain the high-frequency information in SAR images while reducing the speckle noise on high-frequency components, then to recover the high-frequency details in pseudo-optical images.
3. A CFRWD-GAN (cross-fusion reasoning and wavelet decomposition GAN) is presented in this work, which contains one generator and one discriminator. There are two branches in the generator: one with the CFR structure and the other is the WD branch. Outputs from the two branches are fused by a learnable coefficient. Moreover, adversarial and high-dimensional feature matching loss are used to train the CFRWD-GAN.
4. Extensive experiments have been performed on the SEN1-2 and QXS-SAROPT datasets. The results of experiments display that the pseudo-optical images generated by CFRWD-GAN be not only similar to the ground truth in visual inspection but also achieve excellent evaluation metrics, which demonstrates the advantage of our model over the highly advanced STOIT task.

## 2. Related Work

### 2.1. GANs-Based Image-to-Image Translation

One of the most important developments is the GANs, which was put forward by Goodfellow in 2014. GANs contain two adversarial neural networks, including a generative network, usually represented as G, and a discriminative network, commonly symbolized as D. G tries to trick D by generating a nearly real sample, and D attempts to differentiate between authentic and counterfeit samples. The structures of G and D are not fixed, but G is required to acquire the distribution of the image statistics, and D can classify the output by extracting features. GANs are widely utilized in ITIT. Based on the framework of generative confrontation, Isola et al. put forward a pix2pix method, which successfully realized the translation of segmentation labels to real scenes, contour map to the real object, etc.; Zhu et al. proposed the cycleGAN with two generators and two discriminators to perform style transfer, object conversion, and seasonal conversion in an unsupervised way. Furthermore, to achieve a high-resolution ITIT, Wang et al. [14] suggested the pix2pixHD with a coarse-to-fine generator. Thereafter, the new unsupervised conditional GAN, named NICEGAN [15], has been proposed, which achieves perfect effects in ITIT by reusing a discriminator for encoding. Moreover, many variants [16–18] have been proposed based on the above models.

### 2.2. SAR-to-Optical Image Translation

The STOIT is a branch of ITIT. Several researchers have migrated the general image translation methods to the field of SAR and optical conversion. For instance, Enomoto et al. [19] used the cGAN to complete the conversion of SAR to optical images. Toriya et al. [20] applied the pix2pix to achieve modal unification of optical and SAR images. Wang et al. [21] added supervised information to the cycleGAN model to complete the translation of SAR to optical images. Li et al. [22] applied NICEGAN to convert SAR images to optical images. Sebastianelli et al. [23] applied the STOIT method to complete the cloud removal. Additionally, scholars improved the structure of the generator or discriminator. Kento et al. increased a parallel regional classification network to the generator [24], providing more feature information. Javier et al. applied the atrous spatial pyramid pooling (ASPP) to the U-Net structure [25], expanding the receptive field without adding computational complexity. Guo et al. [26] added a gradient branch to the feature reasoning structure, taking full advantage of the gradient information in SAR images. The above modifications are all based on the reasoning structure of the U-Net [27] or CN-ResBlocks [28], and experiments demonstrate that these improvements improve the quality of the pseudo-optical images in some ways. However, due to the inherent defects of the U-Net and CN-ResBlocks reasoning structures, some information is lost when translating SAR features into optical features, resulting in the blurred outline of images and unclear details. A reasoning structure different from the U-Net and CN-ResBlocks structures needs to be proposed to solve the above questions.

### 2.3. Application of Wavelet Decomposition in Deep Learning

Wavelet decomposition can capture spatial and frequency information in the signal, which is an efficient method in image processing. With the success of DL, wavelet methods are embedded in neural networks to complete different tasks, including super-resolution reconstruction, style transfer, quality augmentation, image demonstration [29], etc. Chan et al. [30] developed a wavelet autoencoder (WAE) that decomposed the original images into sub-images of two low-resolution channels and incorporated WAE into classified neural networks for joint training. Zhang et al. [31] discussed the problem of low efficiency and high memory in GANs and proposed to apply WD in GAN, specifically referring to using WD to extract high-frequency details of images. Zhang et al. [32] proposed a wavelet transform as a variational auto-encoder to retain structural information during image translation. Li et al. [33] used the generator to learn the map of SAR images to the wavelet features and then reconstructed the grayscale images to optimize the content. George et al. [34] suggested an unsupervised paradigm that utilizes the self-supervised segmentation loss

and the discrimination based on the whole image wavelet components. Inspired by the above studies, we combine WD and neural networks to filter out speckle noise from high-frequency components in SAR images while recovering high-frequency details in pseudo-optical images.

## 3. Method

In this section, the CFRWD-GAN model is first introduced, and then the details of the generator and discriminator are illustrated. Finally, the loss functions are described in detail.

### 3.1. CFRWD-GAN

The global architecture of the CFRWD-GAN is displayed in Figure 2. The CFRWD-GAN contains a generator that generates pseudo-optical images and a discriminator that identifies the real and the fake ones.

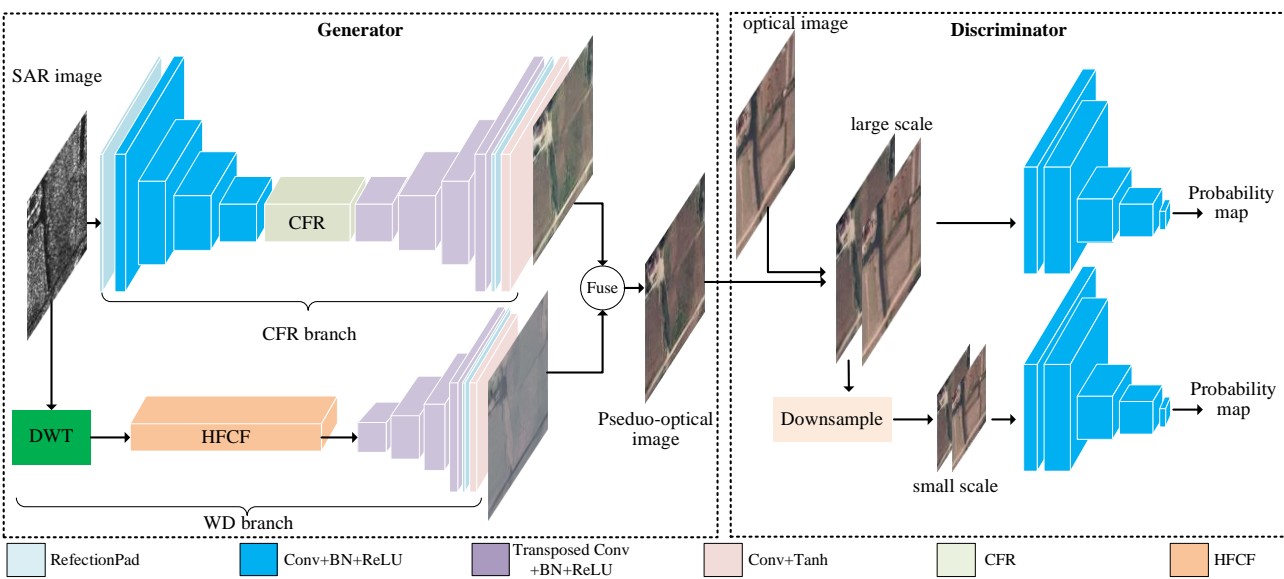

**Figure 2.** An overview of our proposed SAR-to-optical translation (STOI) network.

#### 3.1.1. Generator

The customized version of the generator is built on the pix2pix. Specifically, the U-Net reasoning structure in the pix2pix is replaced with a CFR, and a WD branch is added. The outputs from the branch with a CFR structure and the WD branch are fused with a learnable coefficient. We set the initial fuse coefficient to 1 and find an optimal value through the gradient descent method.

**Cross-fusion reasoning:** In the realm of ITIT, the U-Net structure and the CN-ResBlocks structure are two general reasoning structures in the generation model, as shown in Figure 3. Figure 3a is the U-Net structure used in the pix2pix. In this reasoning structure, feature maps are encoded first and then decoded, and through the skip connection, feature maps in the encoder are combined with the feature maps in the decoder. Although the skip connection retains features in the encoder, the features in the encoder and features in decoders are different in content, and simply concatenating them is not conducive to the translation from SAR features to optical features. Figure 3b is the CN-ResBlocks structure applied in the cycleGAN. The main components in CN-ResBlocks are residual blocks, which solve the optimization problem of the model to some extent and enhance the performance of the algorithm by continuously deepening the network without changing the size of the feature map. However, the feature reasoning is only implemented at the same resolution, failing to complete the feature translation from different scales and leading to the difficulty of generating high-quality images.

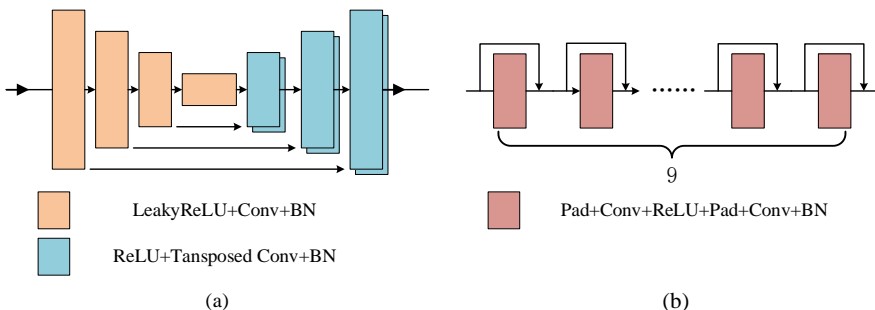

**Figure 3.** Two reasoning structures generally used in translation networks. (**a**) U-Net reasoning structure [24]. (**b**) The cascade of nine residual blocks (CN-ResBlocks) [25].

HRNet [35], which connects feature maps of different resolutions in parallel, and adds interaction between feature maps of different resolutions in the meantime, has gained outstanding results in the area of segmentation, object detection, human pose estimation, etc. Encouraged by the HRNet, this work draws on the idea of high-resolution detailed feature preservation and proposes a new reasoning structure, namely, cross-fusion reasoning structure, to perform the STOIT task. The CFR structure retains high-resolution detail features and low-resolution semantic features simultaneously in the process of feature reasoning. It completes the reasoning of SAR features of different scales to optical image features step by step. In the CFR structure, the whole reasoning process is divided into three stages; each stage will add a new scale branch. Features between different stages through cross-fusion complete the feature transfer, and features of different scales are retained in the reasoning process. CFR effectively solves the problems of information loss and incomplete reasoning by preserving all scales' features in the process of reasoning. In the CFR structure, the horizontal deepening and vertical widening networks are combined to make full use of features of different sizes and simultaneously process high- and low-resolution features in the entire feature reasoning process. As a consequence, better translation performance can be achieved by utilizing the CFR structure.

As shown in Figure 4, there are three stages in the CFR structure, including stage-1, stage-2, and stage-3. Stage-1 consists of two scale branches, namely $N_1^1$ and $N_2^1$. Stage-2 contains three scale branches, denoted as $N_1^2$, $N_2^2$ and $N_2^2$. Stage-3 contains four scale branches, namely $N_1^3$, $N_2^3$, $N_3^3$ and $N_4^3$. Each scale branch is formed with three residual blocks. The input of each branch of the current stage is the result that fully fuses the output of each branch of the previous stage. As shown in Figure 4, $a_1$ represents the input of the CFR structure, which size is $B \times C \times W \times H$. $B$, $C$, $W$ and $H$ denote the batch size, the number of channels of the feature, feature height, and feature width, respectively. $a_2$ with the size of $B \times C \times W/2 \times H/2$ is obtained by downsampling $a_1$. Then, $a_1$ and $a_2$ are input into $N_1^1$ and $N_2^1$, $p_1$ with the size of $B \times C/4 \times W \times H$ and $p_2$ with the size of $B \times C/2 \times W/2 \times H/2$ are outputted from stage-1. Next, $p_1$ and $p_2$ are fused with each other in three scales, and finally, three feature maps, $b_1$ with the size of $B \times C/8 \times W \times H$, $b_2$ with the size of $B \times C/4 \times W/2 \times H/2$ and $b_3$ with the size of $B \times C/2 \times W/4 \times H/4$, are obtained through cross-fusion. The specifical process is expressed by formulas as follows:

$$b_1 = Conv(cat(p_1, U(p_2))) \tag{1}$$

$$b_2 = Conv(cat(D(p_1), p_2)) \tag{2}$$

$$b_3 = Conv(cat(D(D(p_1)), D(p_2))) \tag{3}$$

where the *Conv*, *cat*, *U*, and *D* represent convolution operation, concatenate operation, upsample operation, and downsample operation, respectively; $b_1$, $b_2$, $b_3$, $p_1$, and $p_2$ represent feature maps in different scales and stages. In the second stage, $b_1$, $b_2$, and $b_3$ are input into $N_1^2$, $N_2^2$, and $N_3^2$, respectively, to attain $q_1$ with the size of $B \times C/8 \times W \times H$, $q_2$ with the size of $B \times C/4 \times W/2 \times H/2$, and $q_3$ with the size of $B \times C/2 \times W/4 \times H/4$. Then, similar

operations are performed with $q_1$, $q_2$, and $q_3$ to obtain $c_1$ with the size of $B \times C/16 \times W \times H$, $c_2$ with the size of $B \times C/8 \times W/2 \times H/2$, $c_3$ with the size of $B \times C/4 \times W/4 \times H/4$, and $c_4$ with the size of $B \times C/2 \times W/8 \times H/8$, respectively. The specifical process is expressed by formulas as follows:

$$c_1 = Conv(cat(q_1, U(q_2), U(U(q_3)))) \tag{4}$$

$$c_2 = Conv(cat(D(q_1), q_2, U(q_3))) \tag{5}$$

$$c_3 = Conv(cat(D(D(q_1)), D(q_2), q_3)) \tag{6}$$

$$c_4 = Conv(cat(D(D(D(q_1))), D(D(q_2)), D(q_3))) \tag{7}$$

where the symbols are the same as the above. In the third stage, $c_1$, $c_2$, $c_3$, and $c_4$ are input into $N_1^3$, $N_2^3$, $N_3^3$, and $N_4^3$ to obtain $k_1$ with the size of $B \times C/16 \times W \times H$, $k_2$ with the size of $B \times C/8 \times W/2 \times H/2$, $k_3$ with the size of $B \times C/4 \times W/4 \times H/4$, and $k_4$ with the size of $B \times C/2 \times W/8 \times H/8$, respectively. Finally, $k_1$, $k_2$, $k_3$, and $k_4$ are concatenated in channel dimension after the feature map size is uniform, and through a $1 \times 1$ convolution to integrate channel numbers without changing feature map size. The specifical process is expressed by the formula as follows:

$$d = Conv(cat(D(k_1), k_2, U(k_3), U(U(k_4)))) \tag{8}$$

where $d$ represents the output of the CFR structure; other symbols are the same as the above.

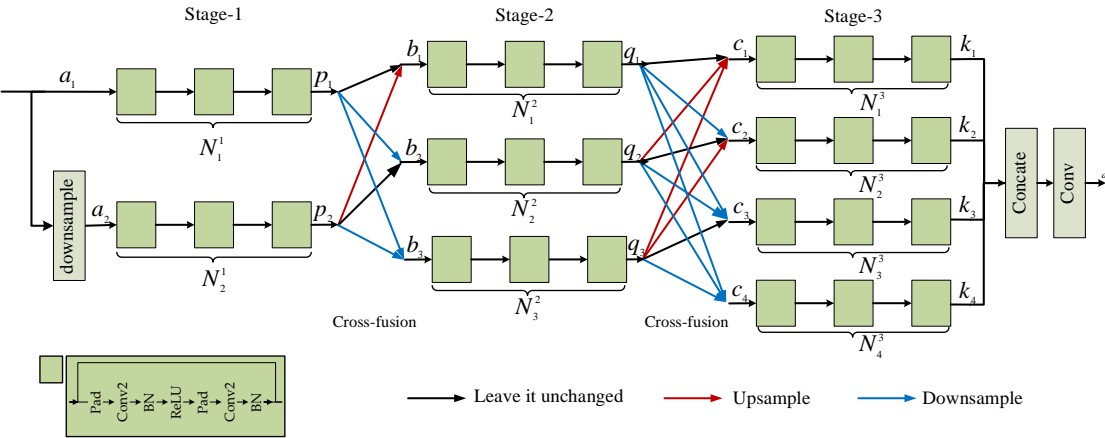

**Figure 4.** The cross-fusion reasoning (CFR) structure in CFRWD-GAN.

From stage-1 to stage-3, the feature information of each scale is preserved, and the high-resolution, detailed features and low-resolution semantic features exist simultaneously throughout the process of feature reasoning.

**Wavelet decomposition branch**: SAR images are heavily scattered with speckle noise, which has a severe impact on the translation performance from STO images. Wavelet decomposition enables the separation of an image into different frequency bands, which makes it possible to independently process different bands. The high-frequency sub-bands of the wavelet coefficients represent the image's edges and details, which are crucial for preserving image structure. The low-frequency bands of the wavelet coefficients refer to smooth regions of the image that contain most of the energy of the image. By thresholding, the high-frequency wavelet coefficients and speckle noise can be removed from the image, while important image details can be preserved. Wavelet decomposition is a common approach to reduce speckle noise in SAR images, mainly due to its ability to decompose an image into different frequency bands and identify noise at different scales.

Considering the application of WD in noise reduction, we construct a parallel branch based on WD to suppress the speckle noise inherent in SAR images. The WD branch

consists of a WD structure, a high-frequency feature coding and filtering (HFCF) structure, and a feature decoding structure. The WD structure is designed to decompose the SAR image into a series of approximate and detailed components, which are then smoothed by the HFCF structure. The output with detailed contour information is achieved through the decoding structure. The specific structure of the WD branch is shown in Figure 5.

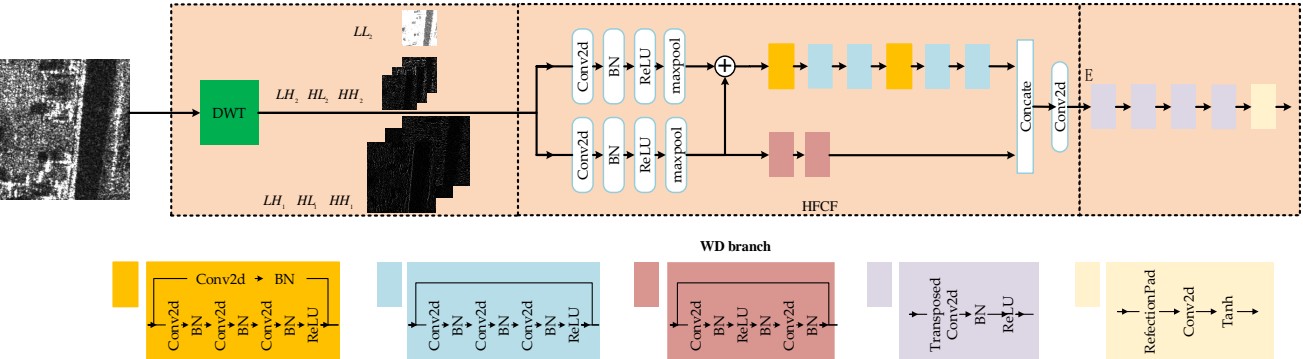

**Figure 5.** The architecture of wavelet decomposition branch.

The first step of the WD branch is to decompose the SAR image into different frequency components. In [36], Yu chose the Haar wavelet function as the basis of the WD. Therefore, the Haar wavelet function is chosen in our paper as the basis of the WD. We set decompose level as 2. Through the first decomposition, four components are obtained, namely $LL_1$, $LH_1$, $HL_1$, and $HH_1$, respectively. Next, the second decomposition continues iteratively on $LL_1$, and produces four sub-components, denoted as $LL_2$, $LH_2$, $HL_2$, and $HH_2$. Finally, there are seven image components produced in WD. An example of the image decomposition result is displayed in Figure 6. Given the input X with the size of $B \times C \times W \times H$, we obtain $LL_2$ with the size of $B \times C \times W/4 \times H/4$, $LH_2$ with the size of $B \times C \times W/4 \times H/4$, $HL_2$ with the size of $B \times C \times W/4 \times H/4$, $HH_2$ with the size of $B \times C \times W/4 \times H/4$, $LH_1$ with the size of $B \times C \times W/2 \times H/2$, $HL_1$ with the size of $B \times C \times W/2 \times H/2$, and $HH_1$ with the size of $B \times C \times W/2 \times H/2$. According to frequency, we divide the seven components into three groups: one group is $LL_2$; the second group is $LH_2$, $HL_2$, and $HH_2$; and the third group is $LH_1$, $HL_1$, and $HH_1$.

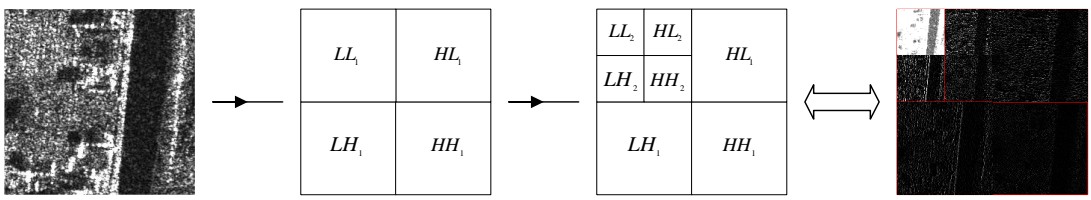

**Figure 6.** The two levels of wavelet decomposition of a SAR image.

To tackle the speckle noise in high-frequency image components, the HFCF structure is proposed to filter noise and recover the optical high-frequency feature in the second and third groups simultaneously. Considering the wide application of the residual network in the image processing field, we design the HFCH structure with the blocks used in ResNet101 and ResNet18. The output of the HFCF structure is the high-frequency optical feature E, and E is input into the decoding structure to obtain the final output.

### 3.1.2. Discriminator

PatchGAN [11] is used as the discriminator in the pix2pix, which has been shown to be effective in improving local detail generation. Therefore, we continue to employ PatchGAN to identify whether an image in each N × N patch is real or fake in our CFRWD-GAN. Furthermore, two sub-branches are constructed in the discriminator to help differentiate the

authenticity of images at various scales. The two sub-branches have an identical structure, as shown in Figure 2.

*3.2. Loss Functions*

The loss function of the pix2pix contains the least squares loss function [37], $L_{LSGAN}$ and the mean absolute loss, $L_1$. $L_{LSGAN}$ is used as an adversarial loss to penalize the generated image distribution away from the real image distribution and improve the stability of the generated images. This paper retains the loss function $L_{LSGAN}$ and uses the high-dimensional feature matching loss $L_{FM}$ to replace the loss function $L_{L1}$. Comparing to $L_1$, $L_{FM}$ can stable the training of the network, which can guide the generator to produce real images from multiple scales [14].

Denote the loss function of our CFRWD-GAN model as $L(G, D)$ especially included two items $L_{LSGAN}$ and $L_{FM}$. The loss function can be formulated by solving the following min-max problem with the function:

$$\min_{G} \max_{D} L(G, D) = L_{LSGAN} + \lambda L_{FM} \tag{9}$$

where $\lambda$ is the hyperparameter adjusting the balance between $L_{LSGAN}$ and $L_{FM}$. In our experiment, $\lambda$ is set to 10. $L_{LSGAN}$ is interpreted as follows:

$$L_{LSGAN} = \frac{1}{2} E_{y \sim P_{data}(y)} [D(Y)]^2 + \frac{1}{2} E_{x \sim P_{data}(x)} [D(G(X)) - 1]^2 \tag{10}$$

where $G$ and $D$ refer to the generator and the discriminator, respectively. $X$ and $Y$ represent SAR images and optical images, respectively. $L_{FM}$ is defined as follows:

$$L_{FM} = \sum_{i=0}^{M} \frac{1}{C_i W_i H_i} \left| D^i(Y) - D^i(G(X)) \right|_1 \tag{11}$$

where $D^i(\cdot)$ represents the *i*-th feature extractor of the discriminator $D$, and $G$ represents the generator; $C_i$, $W_i$, and $H_i$ indicate the *i*-th layer's channel number, width, and height, respectively. $M$ is the number of feature extraction layers.

## 4. Experiments

*4.1. Dataset and Evaluation Metrics*

4.1.1. Dataset

The QXS-SAROPT [38] and SEN1-2 [9] datasets are employed in this study to evaluate the proposed CFRWD-GAN and other comparative methods. The above datasets are mainly used in the field of SAR-to-optical image translation.

QXS-SAROPT dataset is put forward in order to promote the development of deep learning-based SAR-optical fusion approaches. QXS-SAROPT consists of Gaofen-3 images and Google Earth multispectral images, often used for SAR-to-optical image translation and matching. The size of each image is $256 \times 256$ pixels with a spatial resolution of 1 m.

The SEN1-2 dataset collects image patches from across the globe and throughout all meteorological seasons, providing a valuable data source for machine learning researchers working in remote sensing interpretation. The dataset is published by Schmitt et al., which has 282,384 SAR-optical image pairs attained by the Sentinel-1 and Sentinel-2 satellites, including various landforms. Referring to the dataset setting method in published papers [19–21], after removing large areas of sea and images disturbed by clouds, we selected five landscapes images, including suburbs, dense urban areas, farmland, rivers, and hills which are well included in folders S5, S45, S52, S84, and S100. Therefore, S5, S45, S52, S84, and S100 in SEN1-2 are selected in our experiments. Each dataset is divided into a 4:1 ratio to form the training and testing datasets, respectively.

Detailed information on the SAR images in SEN1-2 and QXS-SAROPT datasets is presented in Table 1. SAR images in the SEN1-2 dataset are vertically polarized (VV) data

acquired with the Sentinel-1 satellite, which is equipped with a C-band SAR sensor. The SAR operation mode is strip map. For SAR images in the QXS-SAROPT dataset, they are single-polarized data acquired with the Gaofen-3 satellite equipped with a C-band SAR sensor. The SAR operation mode is spotlight.

**Table 1.** Detailed information on the SAR images in SEN1-2 and QXS-SAROPT datasets.

| Dataset | Resolution (m) | Polarization | Frequency Band | Operation Mode |
|---------|----------------|--------------|----------------|----------------|
| SEN1-2 | $10 \times 10$ | VV | C-band | stripmap |
| QXS-SAROPT | $1 \times 1$ | VV/HH | C-band | spotlight |

4.1.2. Evaluation Methods

The effectiveness of STOIT is evaluated based on four metrics, root mean square error (RMSE) [39], peak signal-to-noise ratio (PSNR) [39], structural similarity index (SSIM) [40], learned perceptual image patch similarity (LPIPS) [41].

**RMSE**: The RMSE mainly judges the pixel deviation between the ground truth and the generated image. The lower the value of RMSE, the higher the similarity between the two images. The RMSE calculation formula is shown as follows:

$$RMSE(y, y^*) = \sqrt{\frac{1}{WH} \sum_{m=0}^{W-1} \sum_{n=0}^{H-1} [y(m,n) - y^*(m,n)]^2} \tag{12}$$

where $y$, $y^*$, $W$, and $H$ represent the real optical image, the pseudo-optical image, the image width, and the image height; $(m,n)$ represents pixel location in the image.

**PSNR**: The PSNR uses the local mean error to judge the difference between the two images. Higher PSNR values indicate smaller image distortion and more similarity between the two images. The PSNR calculation formula is as shown follows:

$$PSNR(y, y^*) = 10 \log \frac{255^2}{MSE} = 10 \log \left[ \frac{255^2 WH}{\sum\limits_{m=0}^{W-1} \sum\limits_{n=0}^{H-1} [y(m,n) - y^*(m,n)]^2} \right] (dB) \tag{13}$$

where the symbols as the same as above.

**SSIM**: The SSIM measures the similarity between the two images from three aspects: brightness, contrast, and structure. The SSIM values range is [0, 1], and the closer the value is to 1, the higher the similarity between the two images. The calculation formula is as shown follows:

$$SSIM(y, y^*) = [a(y, y^*)]^\alpha \cdot [b(y, y^*)]^\beta \cdot [c(y, y^*)]^\gamma \tag{14}$$

where,

$$a(y, y^*) = \frac{2\mu_y \mu_{y^*} + A_1}{\mu_y^2 + \mu_{y^*}^2 + A_1} \tag{15}$$

$$b(y, y^*) = \frac{2\sigma_y \sigma_{y^*} + A_2}{\sigma_y^2 + \sigma_{y^*}^2 + A_2} \tag{16}$$

$$c(y, y^*) = \frac{\sigma_{yy^*} + A_3}{\sigma_y \sigma_{y^*} + A_3} \tag{17}$$

where $a$, $b$ and $c$ refer to brightness comparison, contrast comparison, and structure comparison, respectively; $\alpha$, $\beta$, and $\gamma$ are generally set to 1; $\mu_y$, $\mu_{y^*}$, $\sigma_y^2$, $\sigma_{y^*}^2$ and $\sigma_{yy^*}$ represent the mean of $y$, the mean of $y^*$, the variance of $y$, the variance of $y^*$, and the covariance of $y$ and $y^*$; $A_1$, $A_2$, and $A_3$ are constants, which are set to 1.

**LPIPS**: Contrary to conventional methods, the LPIPS is more in line with human perception. The lower the LPIPS means, the more similarity between the two images. The definition of LPIPS is represented as follows:

$$LPIPS(y, y^*) = \sum_l \frac{1}{H_l W_l} \sum_{h,w} \left|\left| w_l \odot (a_{hw}^l - a^{*l}_{hw}) \right|\right|_2^2 \tag{18}$$

where $a_{(hw)}^l$ and $a^{*l}_{(hw)}$ denote the feature maps of $y$ and $y^*$, respectively, in the $l$-th layer of the assessment network; $w_l$ denotes the computation of the cosine distance between feature maps.

### 4.2. Training Details

Experiments in this work are conducted in PyTorch and on one single GPU of RTX 2080 Ti. The total epoch and batchsize sizes are set to 200 and 1, respectively. Both the generator and the discriminator used the Adam solver with $\beta_1 = 0.5$ and $\beta_2 = 0.999$. All networks are trained with a fixed learning rate of $\lambda = 0.0002$ for the first 100 epochs and then linearly decreases to 0 over the following 100 epochs.

### 4.3. Ablation Study

**Wavelet decomposition level:** To determine the effect of wavelet decomposition levels on translation performance, we set different wavelet decomposition levels and explore their impact on translation performance. Experiments are conducted on the SEN1-2 dataset, and the results are shown in Table 2.

**Table 2.** Translation performance of different wavelet factorization series in SEN1-2. The rising arrow indicates that the higher the value of this item, the better the performance. The down arrow indicates that the smaller the value of this item, the better the performance.

| Dataset | Level | RMSE ↓ | PSNR ↑ | SSIM ↑ | LPIPS ↓ |
|---------|-------|--------|--------|--------|---------|
| SEN1-2  | 2     | 31.9840 | 18.9681 | 0.5619 | 0.3994 |
|         | 3     | 35.0578 | 18.5980 | 0.5588 | 0.4131 |
|         | 4     | 33.8520 | 18.4787 | 0.5550 | 0.4019 |
|         | 5     | 33.1012 | 18.6235 | 0.5637 | 0.4115 |

With the increase of wavelet decomposition levels, the performance of the network for SAR-to-optical image translation is not improved. The values of RMSE and LPIPS between the generated pseudo-optical images and the real optical images increase, while PSNR and SSIM decrease in value as the number of decomposition levels increases. Additionally, because the number of decomposition levels increases, the training time of the network increases accordingly. Therefore, we choose the 2-level wavelet in our CFRWD-GAN model.

**Number of discriminator branches:** To determine the impact of different numbers of discriminator branches on translation performance, we set 2, 3, 4, and 5 discriminator branches in the CFRWD-GAN model, respectively. Experiments are performed on the SEN1-2 dataset, and the results are shown in Table 3.

**Table 3.** Translation performance of different numbers of discriminator branches. The rising arrow indicates that the higher the value of this item, the better the performance. The down arrow indicates that the smaller the value of this item, the better the performance. The best results are shown in boldface.

| Dataset | Number | RMSE ↓ | PSNR ↑ | SSIM ↑ | LPIPS ↓ |
|---------|--------|--------|--------|--------|---------|
| SEN1-2  | 2      | **31.9840** | **18.9681** | **0.5619** | **0.3994** |
|         | 3      | 36.9297 | 18.2731 | 0.5101 | 0.4453 |
|         | 4      | 37.6370 | 18.4806 | 0.5095 | 0.4282 |
|         | 5      | 34.2404 | 18.3739 | 0.4859 | 0.4364 |

As the number of discriminator branches increases, the performance of CFRWD-GAN for SAR-to-optical image translation does not improve. When the number of discriminator branches is set to 2, the RMSE and LPIPS between pseudo-optical and real-optical images are the lowest among different numbers of discriminator branches. As the number of discriminator branches increases, the values of PSNR and SSIM decrease, which means that the translation performance becomes worsens. Therefore, we set the number of discriminator branches to 2 in the CFRWD-GAN model.

To confirm the validity of the CFRWD-GAN, experimentations are conducted with different conditions set on the S5, S45, S52, S84, and S100 datasets under the SEN1-2 spring folder.

**Cross-fusion reasoning structure:** In order to translate different scales of SAR features to optical features, the CFR structure is presented to improve the output of the reasoning structure. The assessments are displayed in Table 4. Compared with the U-Net reasoning structure and CN-ResBlocks reasoning structure, CFR provides PSNR of 4.7059, 4.1901, and 1.1917, 0.9898 improvements and achieves RMSE of 35.8079 and 45.5217, PSNR of 17.7668 and 15.2796, SSIM of 0.4083 and 0.2182, the LPIPS of 0.4739 and 0.5679, respectively, on dataset S5 and S45. The presentation of generated images on SEN1-2 is exhibited in Figure 7. The particulars are marked with red boxes and magnified for presentation. In Figure 7, column (a) are SAR images, from column (b) to column (d) are the images generated by the network with U-Net reasoning structure, the CN-ResBlocks reasoning structure, and the CFR structure, respectively. Column (e) are real optical images. As depicted in Figure 7, images generated with the CFR structure have distinct outlines, clearer edges, and more exact details than images generated by other reasoning structures. In the first row, the residential area in column (d) is well translated compared with those in column (b) and column (c) and more similar to the optical image. In the second row, the edge of the lake in column (d) is clearer than those in column (b) and column (c). Moreover, in the third row, compared with column (b) and column (c), the road is translated perfectly in column (d). In the fourth row, from column (b) to column (d), the green part in the red box is gradually similar to the real one. In order to confirm that the CFR structure retains more feature information during the feature reasoning stage, we visualize the feature maps output from the CFR structure and the features inferred in the U-Net reasoning structure and CN-ResBlock reasoning structure. The visualization is displayed in Figure 8. Figure 8a–e is the feature maps output from the U-Net structure, the CN-ResBlock structure, and the CFR structure. According to Figure 8, we can see that more feature information is clearly preserved in the CFR structure than in the other reasoning structures. In Figure 8b, the reasoned features suffer many interferences from the encoding stage because of the direct connection between features in the coding stage and the features in the decoding stage. Compared with the reasoned features in Figure 8d, the reasoned features in Figure 8c lost other scale information because the feature reasoning was carried out at one scale. The reasoned features obtained by the CFR structure do not interfere with the encoding stage features. Additionally, in the reasoning time, different scale features are fused with each other avoiding incomplete feature reasoning. From these details, it is noticeable that the CFR structure can effectively retain more information than the other two reasoning structures and achieve better translation performance.

**Table 4.** Comparisons of various reasoning structures of our network on SEN1-2 spring S5, S45, S52, and S100 datasets. The rising arrow indicates that the higher the value of this item, the better the performance. The down arrow indicates that the smaller the value of this item, the better the performance. The best results are shown in boldface.

| SEN1-2 | Versions | RMSE ↓ | PSNR ↑ | SSIM ↑ | LPIPS ↓ |
|--------|----------|--------|--------|--------|---------|
| S5 | baseline + U-Net | 59.2862 | 13.0609 | 0.1302 | 0.5947 |
| | baseline + CN-ResBlocks | 55.4026 | 13.5767 | 0.1570 | 0.5804 |
| | baseline + CFR | **35.8079** | **17.7668** | **0.4083** | **0.4739** |
| S45 | baseline + U-Net | 52.1455 | 14.0879 | 0.0969 | 0.6053 |
| | baseline + CN-ResBlocks | 51.0880 | 14.2898 | 0.1099 | 0.5968 |
| | baseline + CFR | **45.5217** | **15.2796** | **0.2182** | **0.5679** |
| S52 | baseline + U-Net | 53.2916 | 14.1577 | 0.1492 | 0.6296 |
| | baseline + CN-ResBlocks | 48.4358 | 14.9321 | 0.1852 | 0.6091 |
| | baseline + CFR | **31.5664** | **18.8666** | **0.4038** | **0.5092** |
| S84 | baseline + U-Net | 53.6439 | 13.7390 | 0.0960 | 0.5911 |
| | baseline + CN-ResBlocks | 43.9750 | 15.4585 | 0.1372 | 0.5670 |
| | baseline + CFR | **34.0315** | **17.9611** | **0.3642** | **0.5130** |
| S100 | baseline + U-Net | 37.3050 | 17.1164 | 0.3216 | 0.5777 |
| | baseline + CN-ResBlocks | 34.3218 | 17.9180 | 0.3828 | 0.5399 |
| | baseline + CFR | **26.6837** | **20.9478** | **0.4726** | **0.5047** |

**Wavelet Decomposition branch:** In order to efficiently separate the frequency information and filter the noise contained in the high-frequency components, the wavelet decomposition branch is presented to enhance the details of the pseudo-optical image. The result of the quantitative evaluation is shown in Table 5. Compared with the network without WD, the WD branch provides RMSE of 2.2659, PSNR of 0.3113, SSIM of 0.0591 and LPIPS of 0.0101 improvements and achieves RMSE of 33.5420, PSNR of 18.0781, SSIM of 0.4674, and LPIPS of 0.4638 on dataset S5. On dataset S84, the WD branch provides RMSE of 0.9183, PSNR of 0.2520, SSIM of 0.0473, and LPIPS of 0.0025 improvements and achieves RMSE of 33.1132, PSNR of 18.2131, SSIM of 0.4115, and LPIPS of 0.5105. It is obvious that the WD branch can effectively separate SAR images into muti-frequency components and filter the noise contained in high-frequency components of SAR images, and achieves better translation performance. The presentation of generated images on SEN1-2 is exhibited in Figure 9. In Figure 9, column (a) are SAR images, column (b) are images generated with the CFR structure, column (c) are images generated with the CFR structure and the WD branch, and column (f) are real optical images. As shown in Figure 9, with the addition of the WD branch, the faint details in the SAR image translate well to the details in the pseudo-optical image. For example, in the first row of Figure 9, the white road in the SAR image is covered with speckle noise and cannot be discerned by human eyes, while the white road is well translated in column (c). As shown in Figure 9, the WD branch can separate the speckle noise and help recover the blurred part of SAR images to clear pseudo-optical images.

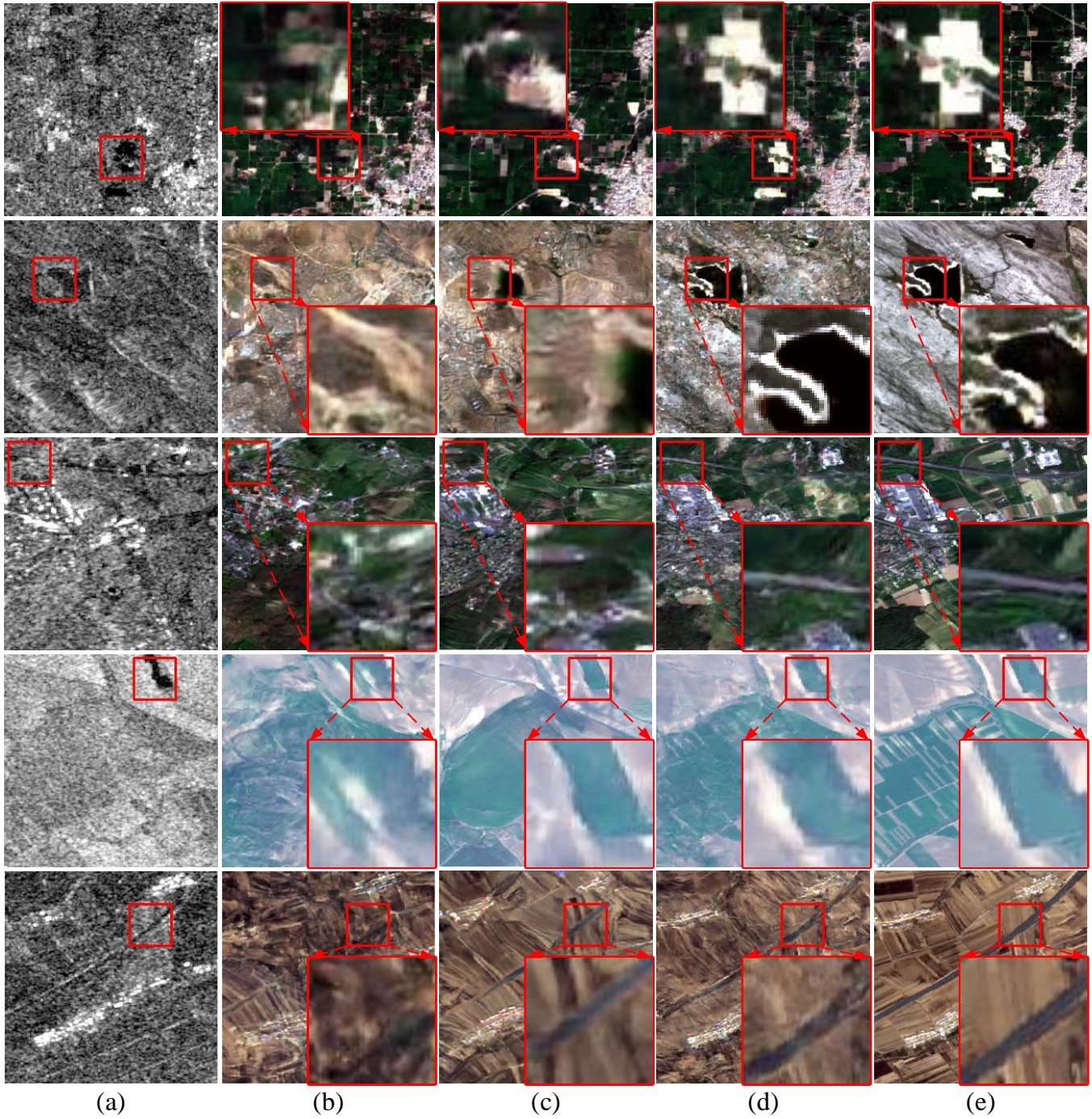

**Figure 7.** The representation of pseudo-optical images generated with different reasoning structures. The particulars are marked with red boxes and magnified for presentation. (**a**) SAR images. (**b**) Pseudo-optical images generated with U-Net reasoning structure. (**c**) Pseudo-optical images generated with CN-ResBlocks reasoning structure. (**d**) Pseudo-optical images generated with CFR structure. (**e**) Optical images.

The output from the WD branch is exhibited in Figure 10. Figure 10a–e are SAR images, the outputs of the WD branch, the outputs of the branch based on CFR structure, the fusion results of the WD branch, and the branch based on CFR structure. Figure 10e is the real optical image. As shown in Figure 10, the high-frequency features in the SAR images are preserved and translated into optical features by the WD branch. In Figure 10a, the high-frequency details are disturbed by speckle noise and unclearly present in SAR images, resulting in vague object edges in Figure 10b. However, the WD branch is able to

separate high-frequency features from SAR images containing speckle noise. As shown in Figure 10c, the WD branch outputs distinct object edges. Combining the results of CFR and WD branches, the pseudo-optical images with clear edges are generated, as shown in Figure 10d, which are close to real optical images.

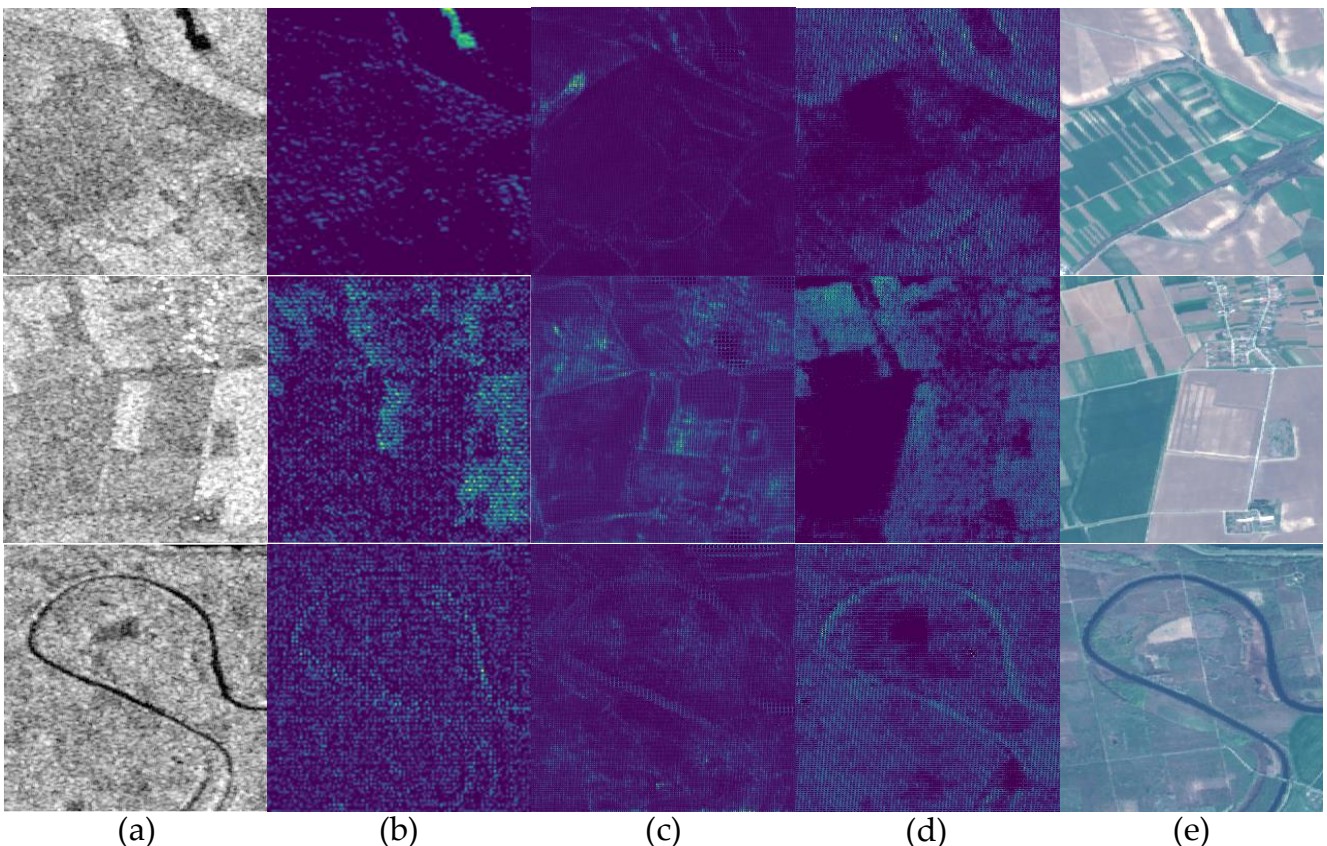

(a)  (b)  (c)  (d)  (e)

**Figure 8.** The visualization of feature maps output from different reasoning structures. (**a**) SAR images. (**b**) The feature maps output from the U-Net reasoning structure. (**c**) The feature maps output from CN-ResBlock reasoning structure. (**d**) The feature maps output from CFR. (**e**) Optical images.

**Two sub-branches discriminator:** In order to distinguish the images from different scales, two sub-branches are constructed in the discriminator. The result of the quantitative metric is shown in Table 5. Compared with single scale discriminator, two sub-branches discriminator provides RMSE of 0.7298, PSNR of 0.3964, SSIM of 0.0739, and LPIPS of 0.0178 improvements and achieves RMSE of 32.3843, PSNR of 18.6095, SSIM of 0.4854, and LPIPS of 0.4927 on dataset S84. The assessment of five datasets shows that two sub-branches discriminator improves the quality of the pseudo-optical images. In Figure 9c, the WD branch recovers detailed edge information; however, the width of edges is still thick, which might be caused by the discriminator that only authenticates the image on one scale. After adding the two sub-branch discriminators, the edges become thinner and close to the edges in real optical images.

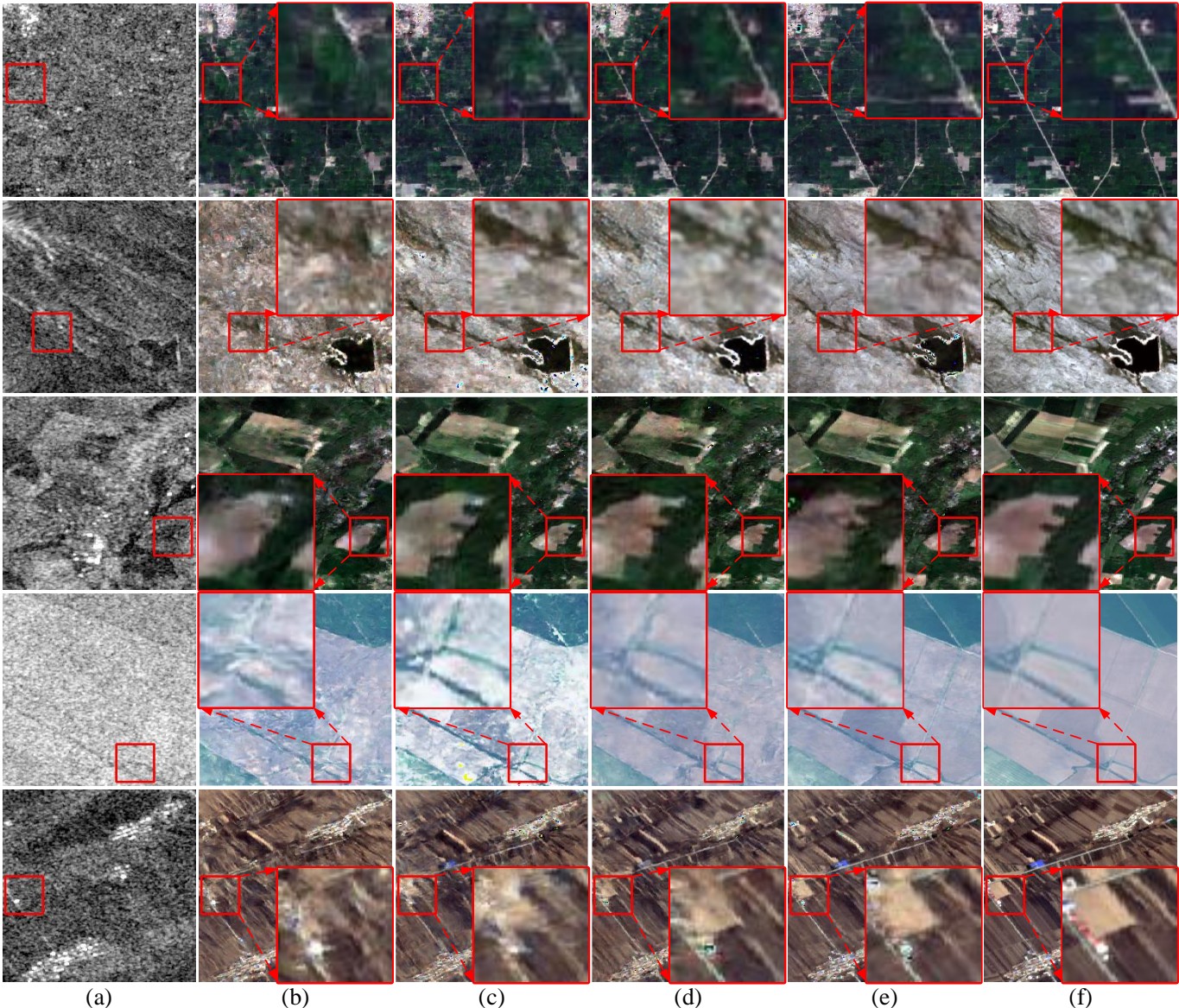

(a)　　　　　(b)　　　　　(c)　　　　　(d)　　　　　(e)　　　　　(f)

**Figure 9.** Examples of results of ablation study. The particulars are marked with red boxes and magnified for presentation. (**a**) SAR images. (**b**) Baseline + CFR. (**c**) Baseline + CFR + WD. (**d**) Baseline + CFR + WD + 2D. (**e**) Baseline + CFR + WD + 2D + $L_{FM}$. (**f**) Optical images.

$L_{FM}$ **loss function:** In order to guide the generating of images on different scales and stable the training of the network, $L_{FM}$ loss function is used in CFRWD-GAN to enhance the output of the network. The result of the quantitative evaluation is displayed in Table 5. In Table 5, the pseudo-optical images generated with $L_{FM}$ loss have increased SSIM value and reduced LPIPS value compared without $L_{FM}$ loss. Specifically, compared without $L_{FM}$ used in training, $L_{FM}$ provides RMSE of 1.2182, PSNR of 0.6336, SSIM of 0.1049, and LPIPS of 0.0802 improvements and achieves RMSE of 31.1652, PSNR of 19.2431, SSIM of 0.5903, and LPIPS of 0.4125 on dataset S84. Qualitative representation is displayed in Figure 9. Part particulars are annotated with red boxes and magnified for a clear presentation. The results in Figure 9e show that $L_{FM}$ loss works well on the texture restoration of pseudo-optical images. In Figure 9e, the texture and details in pseudo-optical images are clearer than those in Figure 9d. $L_{FM}$ loss helps the CFRWD-GAN model generate pseudo-optical images from different levels of features, and finally, pseudo-optical images are gained with true texture information and clear edge information.

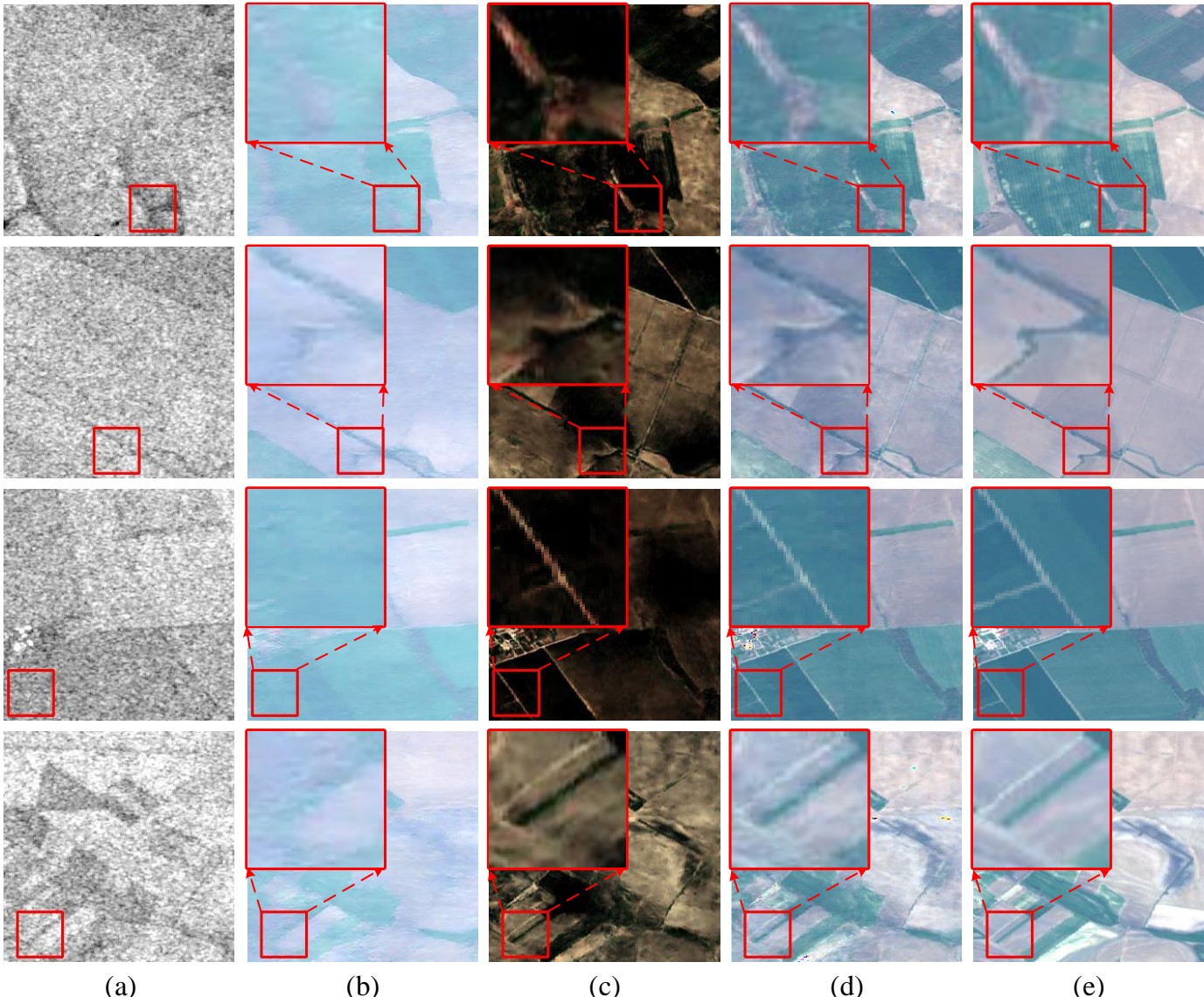

(a)       (b)       (c)       (d)       (e)

**Figure 10.** The representation of the outputs of the WD branch. The particulars are marked with red boxes and magnified for presentation. (**a**) SAR images. (**b**) The output of the branch based on the CFR structure. (**c**) The output of the WD branch. (**d**) The fusion results of the WD branch and the branch based on the CFR structure. (**e**) Optical images.

### 4.4. Comparison Experiments

To validate the effectiveness of the CFRWD-GAN, comparative experiments with pix2pix [11], cycleGAN [12], S-cycleGAN [21], NICEGAN [15], and GANILILA [42] are represented on the QXS-SAROPT and SEN1-2. The quantitative comparison results are displayed in Tables 6 and 7, and the best results are marked in boldface. In QXS-SAROPT, the CFRWD-GAN attains the highest values of the PSNR and SSIM among the six models. Regarding LPIPS, the value is the lowest among the six methods, indicating that the pseudo-optical images generated by CFRWD-GAN are better and similar to the ground truth. Regarding SSIM, Our CFR-GAN improved by 30.01%, 27.39%, 17.13%, 34.76%, and 193.60% compared with pix2pix, cycleGAN, S-cycleGAN, NICEGAN, and GANILLA, respectively.

**Table 5.** Comparisons of ablation study of our network on SEN1-2 S5, S45, S52, S84, and S100 datasets. CFR: cross-fusion reasoning structure, WD: wavelet decomposition, 2D: two sub-branches discriminator, $L_{FM}$: the high-dimensional feature matching loss. The rising arrow indicates that the higher the value of this item, the better the performance. The down arrow indicates that the smaller the value of this item, the better the performance. The best results are shown in boldface.

| SEN1-2 | CFR | WD | 2D | $L_{FM}$ | RMSE ↓ | PSNR ↑ | SSIM ↑ | LPIPS ↓ |
|---|---|---|---|---|---|---|---|---|
| | √ | | | | 35.8079 | 17.7668 | 0.4083 | 0.4739 |
| S5 | √ | √ | | | 33.5420 | 18.0781 | 0.4674 | 0.4638 |
| | √ | √ | √ | | 33.2064 | 18.3026 | 0.5098 | 0.4891 |
| | √ | √ | √ | √ | **31.9840** | **18.9681** | **0.5619** | **0.3994** |
| | √ | | | | 45.5217 | 15.2796 | 0.2182 | 0.5679 |
| S45 | √ | √ | | | 40.6611 | 16.8970 | 0.3791 | 0.5379 |
| | √ | √ | √ | | 37.3188 | **17.8494** | 0.4168 | 0.5649 |
| | √ | √ | √ | √ | **37.1655** | 17.7945 | **0.4884** | **0.4831** |
| | √ | | | | 31.5664 | 18.8666 | 0.4038 | 0.5092 |
| S52 | √ | √ | | | 31.5834 | 18.9264 | 0.4928 | 0.4928 |
| | √ | √ | √ | | 30.6756 | 19.0305 | 0.5196 | 0.4816 |
| | √ | √ | √ | √ | **29.1952** | **19.5607** | **0.5414** | **0.4450** |
| | √ | | | | 34.0315 | 17.9611 | 0.3642 | 0.5130 |
| S84 | √ | √ | | | 33.1132 | 18.2131 | 0.4115 | 0.5105 |
| | √ | √ | √ | | 32.3843 | 18.6095 | 0.4854 | 0.4927 |
| | √ | √ | √ | √ | **31.1652** | **19.2431** | **0.5903** | **0.4125** |
| | √ | | | | 26.6837 | 20.9478 | 0.4726 | 0.5047 |
| S100 | √ | √ | | | 26.1237 | 21.5959 | 0.6157 | 0.4339 |
| | √ | √ | √ | | **21.6153** | **22.3971** | 0.6002 | 0.4596 |
| | √ | √ | √ | √ | 24.7660 | 22.2410 | **0.7168** | **0.3639** |

**Table 6.** IQA (Image Quality Assessment) with different translation methods on QXS-SAROPT. The rising arrow indicates that the higher the value of this item, the better the performance. The down arrow indicates that the smaller the value of this item, the better the performance. The best results are shown in boldface.

| Dataset | Models/IQA | RMSE ↓ | PSNR ↑ | SSIM ↑ | LPIPS ↓ |
|---|---|---|---|---|---|
| | Pix2pix | **31.3971** | 18.3803 | 0.3602 | 0.6473 |
| | CycleGAN | 37.6152 | 17.1474 | 0.3676 | 0.5753 |
| QXS-SAROPT | S-cycleGAN | 32.1123 | 18.1896 | 0.3998 | 0.6584 |
| | NICEGAN | 47.9546 | 14.6595 | 0.3475 | 0.5859 |
| | GANILLA | 38.9487 | 16.4145 | 0.1595 | 0.6227 |
| | CFRWD-GAN | 33.7660 | **19.0178** | **0.4683** | **0.5238** |

**Table 7.** IQA with different translation methods on SEN1-2. The rising arrow indicates that the higher the value of this item, the better the performance. The down arrow indicates that the smaller the value of this item, the better the performance. The best results are displayed in boldface.

| Sen1-2 | Models/IQA | RMSE ↓ | PSNR ↑ | SSIM ↑ | LPIPS ↓ |
|--------|------------|--------|--------|--------|---------|
| | Pix2pix | 59.2862 | 13.0609 | 0.1302 | 0.5947 |
| | CycleGAN | 58.5405 | 13.4130 | 0.1750 | 0.5563 |
| S5 | S-cycleGAN | 44.3185 | 15.5888 | 0.2306 | 0.6924 |
| | NICEGAN | 76.9458 | 10.9615 | 0.0960 | 0.6974 |
| | GANILLA | 56.8872 | 13.7428 | 0.1795 | 0.5452 |
| | CFRWD-GAN | **31.9840** | **18.9681** | **0.5619** | **0.3994** |
| | Pix2pix | 52.1455 | 14.0879 | 0.0969 | 0.6053 |
| | CycleGAN | 69.3182 | 11.8628 | 0.0804 | 0.6046 |
| S45 | S-cycleGAN | 40.5645 | 16.3255 | 0.1872 | 0.6855 |
| | NICEGAN | 66.6370 | 12.2907 | 0.0829 | 0.6021 |
| | GANILLA | 65.8261 | 12.3702 | 0.0672 | 0.5932 |
| | CFRWD-GAN | **37.1655** | **17.7945** | **0.4884** | **0.4831** |
| | Pix2pix | 53.2916 | 14.1577 | 0.1492 | 0.6296 |
| | CycleGAN | 55.0609 | 13.7640 | 0.1551 | 0.6130 |
| S52 | S-cycleGAN | **25.2093** | **20.7661** | 0.4677 | 0.6257 |
| | NICEGAN | 54.1179 | 13.8628 | 0.1379 | 0.6689 |
| | GANILLA | 53.3846 | 14.0246 | 0.1721 | 0.6356 |
| | CFRWD-GAN | 29.1952 | 19.5607 | **0.5414** | **0.4450** |
| | Pix2pix | 37.3050 | 17.1164 | 0.3216 | 0.5777 |
| | CycleGAN | 39.8048 | 16.5679 | 0.3807 | 0.5491 |
| S84 | S-cycleGAN | 25.4606 | 20.3833 | 0.4558 | 0.5982 |
| | NICEGAN | 35.5919 | 17.3726 | 0.4059 | 0.5476 |
| | GANILLA | 39.5776 | 16.6620 | 0.4090 | 0.5369 |
| | CFRWD-GAN | **24.7661** | **22.2410** | **0.7168** | **0.3639** |
| | Pix2pix | 53.6440 | 13.7390 | 0.0960 | 0.5911 |
| | CycleGAN | 59.8141 | 12.7815 | 0.1032 | 0.5835 |
| S100 | S-cycleGAN | 46.3288 | 15.1808 | 0.2024 | 0.7375 |
| | NICEGAN | 56.5410 | 13.2690 | 0.1120 | 0.5736 |
| | GANILLA | 63.5996 | 12.3382 | 0.1198 | 0.6093 |
| | CFRWD-GAN | **32.3843** | **18.6095** | **0.4854** | **0.4125** |

In the SEN1-2 dataset, it is undeniable that not all metrics of CFRWD-GAN are optimal among the six methods; however, in terms of SSIM and LPIPS, CFRWD-GAN has achieved the best results, indicating promising translation performance. Our model achieves SSIM of 0.5619, 0.4884, 0.5414, 0.7168, and 0.4854 on S5, S45, S52, S84, and S100, respectively.

Qualitative experimental results on QXS-SAROPT are shown in Figure 11. From column (b) to column (h), the image quality is gradually improving. To compare details of the pseudo-optical images, particulars are marked with red boxes in Figure 11. In the first row of Figure 11, the images generated by CFRWD-GAN have a clearer white road than other methods. Moreover, the image generated by CFRWD-GAN in the second row is almost as clear as the ground truth, and the red house is more distinct than other methods. By observing images of the third row in Figure 11, we find that the proposed CFR-GAN

retains color information better and consistently than other methods. Furthermore, in the fourth row of Figure 11, the texture in the image gained by CFRWD-GAN is clearer than the other five methods.

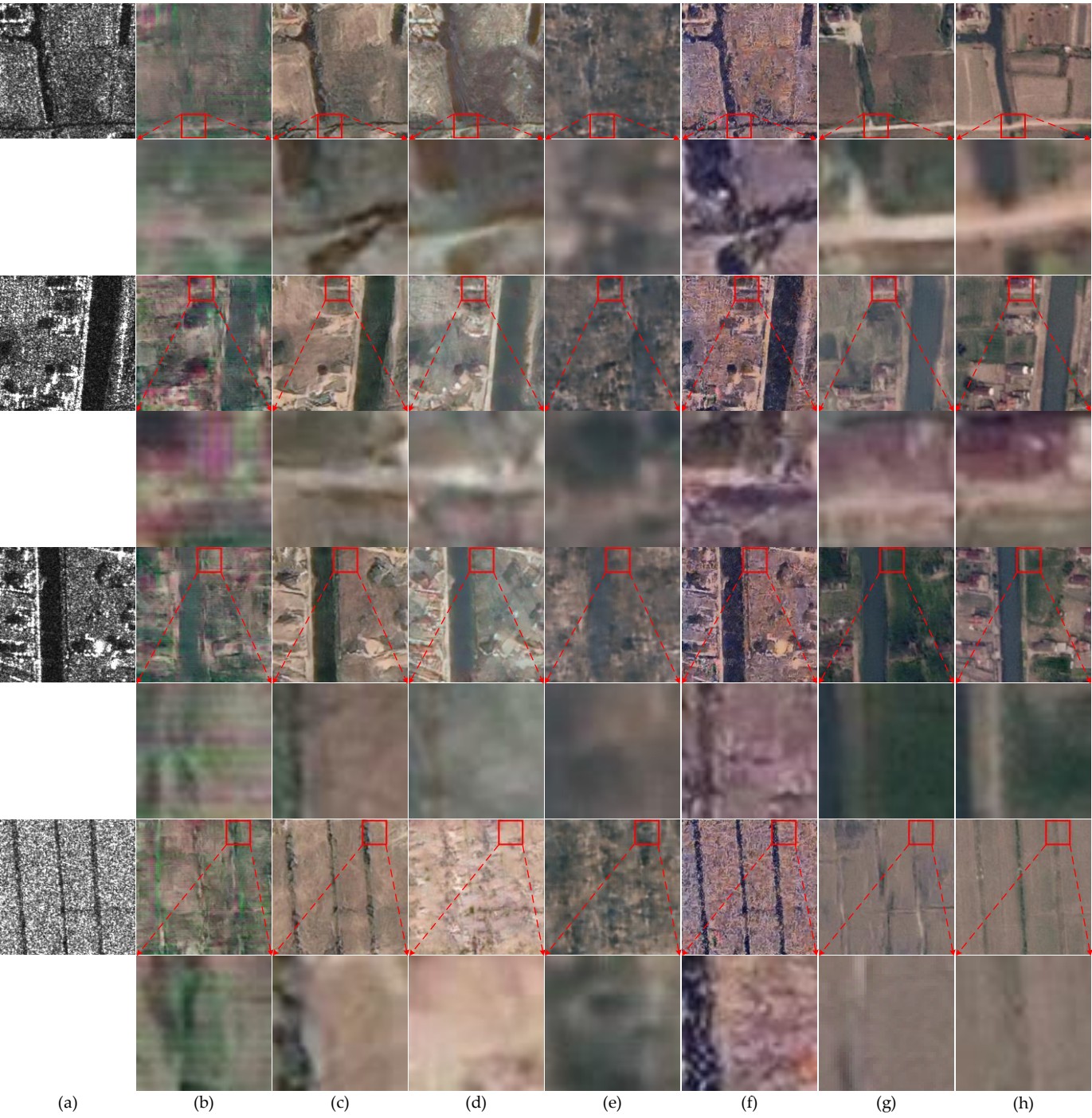

**Figure 11.** Examples of STOIT with different models. The particulars are marked with red boxes and magnified for presentation. (**a**) SAR images. (**b**) pix2pix. (**c**) cycleGAN. (**d**) S-cycleGAN. (**e**) NICEGAN. (**f**) GANILLA. (**g**) CFRWD-GAN. (**h**) Optical images (ground truth).

More examples of pseudo-optical images generated in the SEN1-2 dataset are shown in Figure 12. Some details are marked with red boxes. As shown in Figure 12, our CFRWD-GAN model can generate higher-quality pseudo-optical images than the alternative methods.

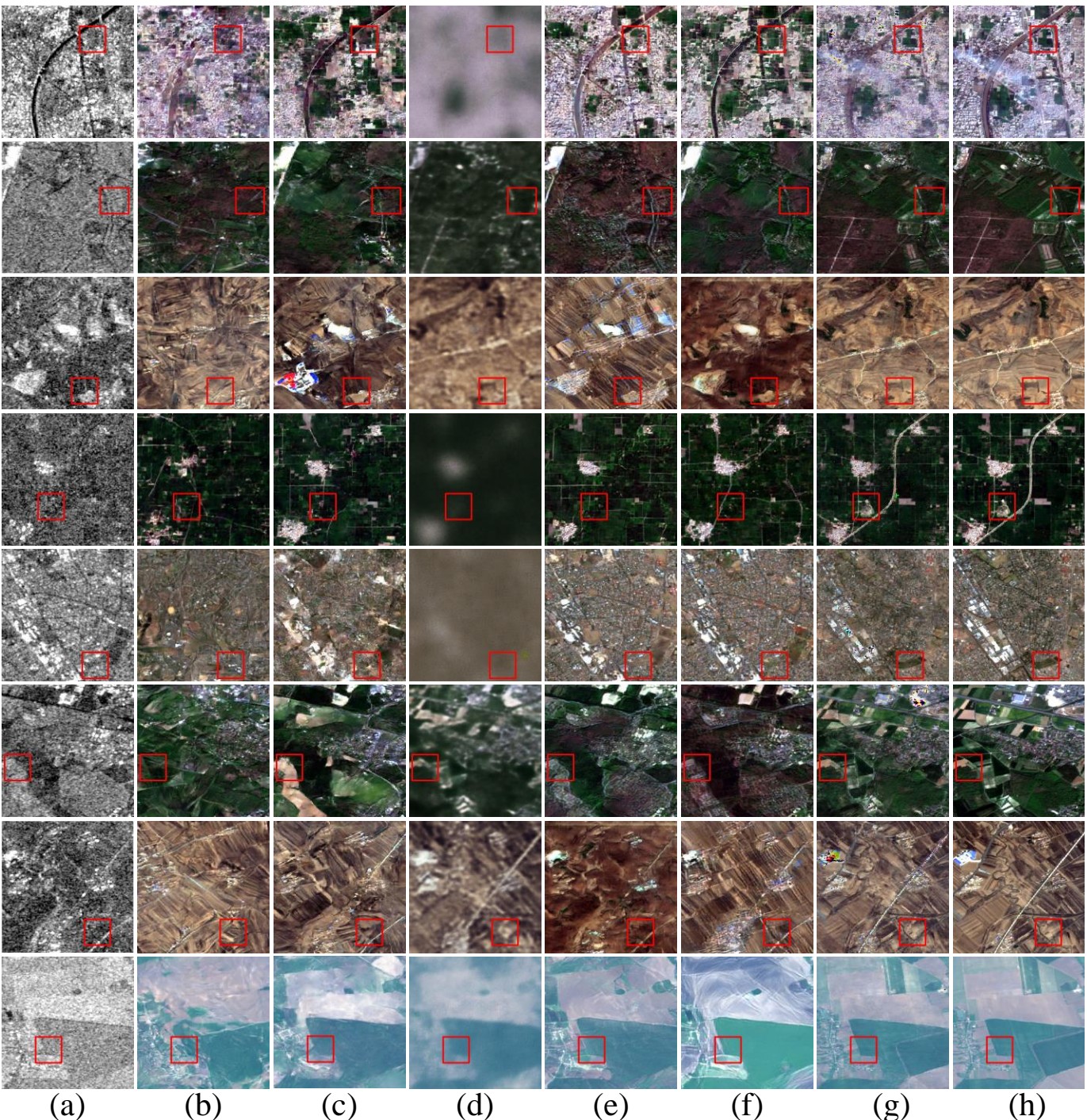

(a)          (b)          (c)          (d)          (e)          (f)          (g)          (h)

**Figure 12.** More examples of pseudo-optical images generated by different models. The particulars are marked with red boxes. (**a**) SAR images. (**b**) pix2pix. (**c**) cycleGAN. (**d**) S-cycleGAN. (**e**) GANILLA. (**f**) NICEGAN. (**g**) CFRWD-GAN. (**h**) Optical images (ground truth).

*4.5. Evaluation of Efficiency*

In addition to assessing the pseudo-optical image quality and visualization outcomes of various methods, we also measure the computational complexity and model parameters in terms of millions of parameters (Params) and floating-point operations per second (FLOPs). All models are computed with an input of $1 \times 256 \times 256$. Table 8 depicts the outcomes. Compared to the other five translation models, our model's performance has significantly improved, despite lacking an advantage in computational complexity. In future projects, we will concentrate on striking a balance between complexity and accuracy.

**Table 8.** Comparison with other networks on FLOPs and Params. G—generator. D—discriminator.

| Model | | FLOPs (G) | Params (M) |
|---|---|---|---|
| Pix2pix | G | 18.152 | 54.414 |
| | D | 3.202 | 2.769 |
| CycleGAN | G | 113.858 | 22.766 |
| | D | 6.304 | 5.532 |
| S-cycleGAN | G | 113.858 | 22.766 |
| | D | 6.304 | 5.532 |
| NICEGAN | G | 67.537 | 16.192 |
| | D | 12.000 | 93.749 |
| GANILLA | G | 18.094 | 7.228 |
| | D | 3.149 | 2.765 |
| CFRWD-GAN | G | 48.631 | 35.800 |
| | D | 4.671 | 5.539 |

## 5. Discussion

Recently, researchers have modified pix2pix and cycleGAN to perform the STOIT task. These improvements include modifications in the structure of the generator and the loss functions.

The commonly used reasoning structures in the generator are the U-Net structure and CN-ResBlocks structure. However, these reasoning structures cause a large amount of information loss. Therefore, we consider whether it is possible to simultaneously retain different levels of features during the reasoning process. Aiming to address the problem of massive information loss during feature reasoning, the CFR, a multi-scale-based structure, is proposed in this paper. The CFR structure retains high-resolution detail features and low-resolution semantic features simultaneously in the process of feature reasoning and completes the reasoning of SAR features of different scales to optical image features step by step. In the CFR structure, the whole reasoning process is divided into three stages; each stage will add a new scale branch. Features between different stages through cross-fusion complete the feature transfer, and features of different scales are retained in the reasoning process. CFR effectively solves the problems of information loss and incomplete reasoning by preserving all scales' features in the process of reasoning. Visualization of the output feature maps of the U-Net, CN-ResBlocks, and CFR structures shows that the CFR structure retains more information than the other two feature reasoning structures.

In addition, to better deal with speckle noise in SAR images and recover high-frequency details of the images, the WD branch is constructed based on wavelet decomposition. Wavelet decomposition enables the separation of an image into different frequency bands, which makes it possible to process different frequency bands independently. The high-frequency sub-bands of the wavelet coefficients represent the edges and details of images, which are crucial for preserving image structure. The low-frequency band of the wavelet coefficients refers to the smoothed regions of the image, which contain most of the image energy. By thresholding the high-frequency wavelet coefficients, speckle noise can be removed from the image while important image details are preserved. Visualization of the WD branch shows that it filters the noise in the wavelet domain and more completely preserves the high-frequency information of the image. Based on the SEN1-2 dataset and the QXS-SAROPT dataset, the effectiveness of the CFR branch and the WD structure is verified by ablation experiments. The qualitative and quantitative comparison results show the superiority of the CFRWD-GAN model.

## 6. Conclusions

In this paper, we present CFRWD-GAN, a novel architecture for supervised STOIT. Driven by the characteristic of the SAR images, we make clear the key problem of the STOIT, including part information loss during the feature stage, making the outline of the

translated pseudo-optical images blurred and semantic information missing, and severe speckle noise exists in the SAR images. To address the above questions, we propose the network specifically to gain better translation results. First, a cross-fusion reasoning structure, which can preserve both high-resolution, detailed features and low-resolution semantic features in the whole process of feature reasoning, is designed in the translation of the SAR image feature to the optical image feature. Second, the wavelet decomposition branch, which contains a wavelet decomposition structure, HFCF structure, and feature decoding structure, is proposed to tackle the high-frequency image components. Finally, the fusion of the WD branch and the branch based on the CFR structure can make the CFRWD-GAN generate high-quality images. In addition, extensive experiments are carried out on SEN1-2 and QXS-SAROPT datasets, and the experimental results show that our method achieves perfect performance in the STOIT task. In the future, we will verify the effectiveness of STOIT for image registration, image detection, image fusion, and other tasks.

**Author Contributions:** Conceptualization, H.Z.; methodology, J.W.; software, J.W.; validation, J.W., H.Z. and L.S.; formal analysis, J.W.; investigation, J.W.; resources, J.W.; data curation, L.S.; writing—original draft preparation, J.W.; writing—review and editing, J.W., H.Z. and L.S.; visualization, X.C.; supervision, S.H., S.L. and Y.Z.; project administration, J.W. All authors have read and agreed to the published version of the manuscript.

**Funding:** This research was funded by the National Natural Science Foundation of China (Grants 62071474).

**Data Availability Statement:** The SEN1-2 dataset used in this study (accessed 19 July 2021) is accessible from https://mediatum.ub.tum.de/1436631. It is shared under the open access license CC-BY. The QXS-SAROPT dataset used in this study is accessible from https://github.com/yaoxu008/QXS-SAROPT (accessed on 1 February 2023).

**Conflicts of Interest:** The authors declare no conflict of interest.

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
