# Peer review of "CFRWD-GAN for SAR-to-Optical Image Translation"

_remotesensing, doi:10.3390/rs15102547_

Round 1

Reviewer 1 Report

Authors present method of CFRWD-GAN for SAR-to-optical image translation in this manuscript. The research is interesting. And comments are listed as follows.

Major issue 1: SAR images presented in this manuscript have one channel but optical images have 3 channels. How could authors transform one channel to 3 channels in this manuscript?

Major issue 2: Authors should present description on reasons of choosing wavelet decomposition for reducing speckle noise. It is uncommon to use wavelet to reduce speckle noise.

Major issue 3:What’s the significance of transforming SAR images to optical images?

Major issue 4: Authors use 2-layers wavelet in this manuscript. How about performances of 3-layers, 4-layers and higher layers wavelet?

Major issue 5: In Figure 11, the partial enlarged region covers part of the result in every sub-figure. And several strong scatters are also covered by partial enlarged region. In this case, the performance of the proposed method is not demonstrated well. It is suggested that authors should not use the partial enlarged region to cover the part of the original result. It is suggested that authors could compare the complete figures and partial enlarged regions separately.

Minor issue 1: Corresponding references of comparison methods should cited when comparison methods are all mentioned.

Minor issue 2: Authors describe that 282384 pairs of data is used in the experiment but only 4 pairs are used for present performance of the proposed method. It is suggested authors should present 8 more different SAR-optical data pairs to further demonstrate the performance of the proposed method.

Reviewer 2 Report

The paper proposes "a cross-fusion reasoning and wavelet decomposition GAN (CFRWD-GAN) for the translation of SAR-to-optical images" The article is technically sound, and its contributions are clearly presented. Also, the proposed method is corroborated by experimental evaluations, and its performance is competitive compared to other methods from the literature. Ablation studies are presented. The paper's readability is good, even with some minor typos and grammar errors. However, its presentation should be improved. The article contains minor problems regarding spaces, notation in equations, and equations formalism. Finally, to improve the paper's quality, the authors should consider:- The motivations and discussion regarding the selection of the CFR and the WD structures. The authors should consider including more related works regarding this topics

- The network parameters and the method's elements should be presented with more details and better discussed.

- Since the focus of the paper is the translation of SAR-to-optical images, more discussion should be presented regarding the SAR characteristics of the data. The presented results are expected for any polarization? Are similar results expected for all frequency bands? For different SAR operation modes?

- The details of the data set should be better presented (even the Sentinel being a quite standard data set) and better discussed. Why the data sets S5, S45, S52, S84, and S100 were selected?

- Overall, more discussion should be provided regarding the results presented in Section 4. Also, Section 5 should be improved overall.

Round 2

Reviewer 1 Report

I think this paper has been revised carefully according to the comments from reviewers. This manuscript can be accepted for publication in this version.